# A PCR-free rapid protocol for one-pot construction of highly diverse genetic libraries

**Michael Woolley**, **Zhilei Chen** *

Microbial Pathogenesis and Immunology Department, Texas A&M Health Science Center, Bryan, Texas, United States of America

* zchen4@tamu.edu

**Data Availability Statement:** All raw image files are included in the submission. Sequencing data and analysis are held in the OSF repository. https://doi.org/10.17605/OSF.IO/2STCW.

## Abstract

*In vitro* protein display methods can access extensive libraries (e.g., $10^{12}$–$10^{14}$) and play an increasingly important role in protein engineering. However, the preparation of large libraries remains a laborious and time-consuming process. Here we report an efficient one-pot ligation & elongation (L&E) method for sizeable synthetic library preparation free of PCR amplification or any purification steps. As a proof of concept, we constructed an ankyrin repeat protein templated synthetic library with $10^{11}$ variants in 150 μL volume. The entire process from the oligos to DNA template ready for transcription is linearly scalable and took merely 90 minutes. We believe this L&E method can significantly simplify the preparation of synthetic libraries and accelerate *in vitro* protein display experiments.

## Introduction

Protein therapeutics have a potentially limitless ability to improve how we treat diseases. However, developing new therapeutic proteins remains a long and challenging process. One of the most effective protein engineering strategies currently is directed evolution. This process mimics the natural evolution and enriches protein variants with the desired functions from large libraries of variants [1, 2]. Since the likelihood of identifying a desired variant is directly proportional to the size of the library, large protein libraries are highly desirable in directed evolution.

A prerequisite of directed evolution is protein display, a process of coupling phenotype (protein function) with genotype (DNA/RNA). Many display technologies have been developed to provide the linkage of phenotype to genotype. The library size afforded by cell-based protein display technologies such as phage, *E. coli*, and yeast display is limited by the cell transformation efficiency and is typically $<10^{10}$, a library size that is relatively straightforward to produce with PCR-based methods [3–9]. Through negation of the transformation step, cell-free protein display technologies such as mRNA [10] and ribosome [11] display can screen much larger libraries with $10^{12}$–$10^{14}$ different variants and are potentially superior to cell-based methods. However, preparation of libraries with $10^{12}$–$10^{14}$ genetic diversity is both time-consuming and laborious [12]. For a gene with 500 bp, a typical PCR reaction (*e.g.*, 50 μL) using 10 ng of DNA template has a maximum diversity of $2\times10^{10}$ (10 ng/500 bp/ 660 (g/ mol/bp) x $6\times10^{23}$ molecules/mol). Thus, a library with $10{12}$–$10^{14}$ variants would require 2.5– 250 mL of PCR reaction. The complexity in library preparation is further compounded in the

**Funding:** Funding provided by a grant ZC received (DP2OD008756) from the National Institutes of Health. The funders had no role in study design, data collection and analysis, decision to publish, or preparation of the manuscript.

**Competing interests:** ZC and MW have jointly submitted a provisional patent (63/416,627) on this technology.

case of antibody or scaffold libraries whose variable regions are scattered throughout the gene, necessitating multiple rounds of PCR reactions followed by gel purification and subsequent ligation of these PCR products to reconstitute the entire gene. For example, Kondo, *et al.* recently reported the synthesis of a monobody library with $10^{13}$ variants that required multiple 15 mL PCR reactions followed by digestion, ligation, and a final amplification in a 60 mL PCR reaction [13, 14]. According to the authors, the entire process took a whopping six months to complete [13].

To simplify the preparation of large libraries for *in vitro* protein display, we here report an alternative Ligation & Extension (L&E) method. To demonstrate this method, we used the Regulatory Factor X-associated Ankyrin-containing protein (RFXANK) [15, 16], a human ankyrin protein (named HARPin here), as a model binder scaffold. HARPin shares high structural homology to the designed ankyrin repeat protein (DARPin) previously developed by Pluckthun and co-workers [17, 18] and consists of 3 ankyrin repeat domains sandwiched between N- and C-capping domains (Fig 1A). The entire HARPin gene has 516 bp. We used a ligation-mediated strategy similar to that described previously [19, 20], which utilizes short antisense splint oligos that anneal at the junctions of adjacent forward strand fragments to generate the full-length gene. Nine synthetic sense oligos spanning the entire length of the HARPin gene and the T7 promoter are synthesized (*i.e.*, NF1-9) and evenly divided into three groups (Fig 1B). The three oligos within each group are first ligated together with the help of short splint oligos (*i.e.*, SP1-8, Fig 1C, step 1). All sense oligos, except for NF1, are phosphorylated at the 5' end, while all splint oligos contain 3' phosphate groups to prevent their ligation. In designing the oligos, we used the following criteria: i) total length <90 nucleotides, ii) splint oligo $T_m$ ~40˚C for the annealing region with each sense oligo, and iii) splint oligos bind the framework region of the binding scaffold and include no randomized codons. Next, the three ligation reactions are combined, and additional splint oligos are added to promote the formation of a long single-stranded DNA representing the entire gene (Fig 1C, step 2). Finally, primer CR, which anneals to the 3' of the target gene, and a polymerase with strong strand displace activity (*i.e.*, *Bst 2.0*), is added to generate the full-length double-stranded gene (Fig 1C, step 3). Although many partial gene fragments exist in the final reaction mixture, only the full-length gene contains the antisense strand of the T7 promoter, enabling the reaction product to be used directly for mRNA synthesis without any purification (Fig 1B, step 4).

To demonstrate the L&E method, a sizeable HARPin library was prepared. In Step 1, three 30 μL ligation reactions, each containing 15 pmoles of the respective sense and splint oligos, were carried out at 37˚C for 30 minutes, resulting in a dominant single-stranded DNA product corresponding to the ligation of the three sense oligos within each group (Fig 2A, yellow arrows). Next, in Step 2, the three ligation reactions were combined along with additional splint oligos in 100 μL reaction volume to generate the single-stranded full-length gene product. After a 30-minute incubation at 37˚C, multiple new bands were visible on the gel. The largest product is presumed to correspond to the ligation product of all the sense oligos (620 nt), which includes the regions for the T7 promoter and ribosome binding site (RBS) (Fig 2A, green arrow). The final ligation reaction also contained multiple smaller DNA products from incomplete ligation of multiple sense oligos. In Step 3, 15 pmoles of primer CR, which contains a region necessary for hybridizing the puromycin-containing oligo, and 48 units of *Bst 2.0*, a polymerase with strong strand displacement activity, were added together with dNTPs. The reaction (150 μL) was carried out at 65˚C for 30 minutes to generate the full-length double-stranded gene (693 bp). Step 3 produced a smear of DNA products ranging from approximately 100 bp to 700 bp (Fig 2B, lane 1). This is unsurprising given that the product of Step 2 contained multiple DNA fragments as well as excess NF oligos that can anneal to each other to yield an array of dsDNA of varying size.

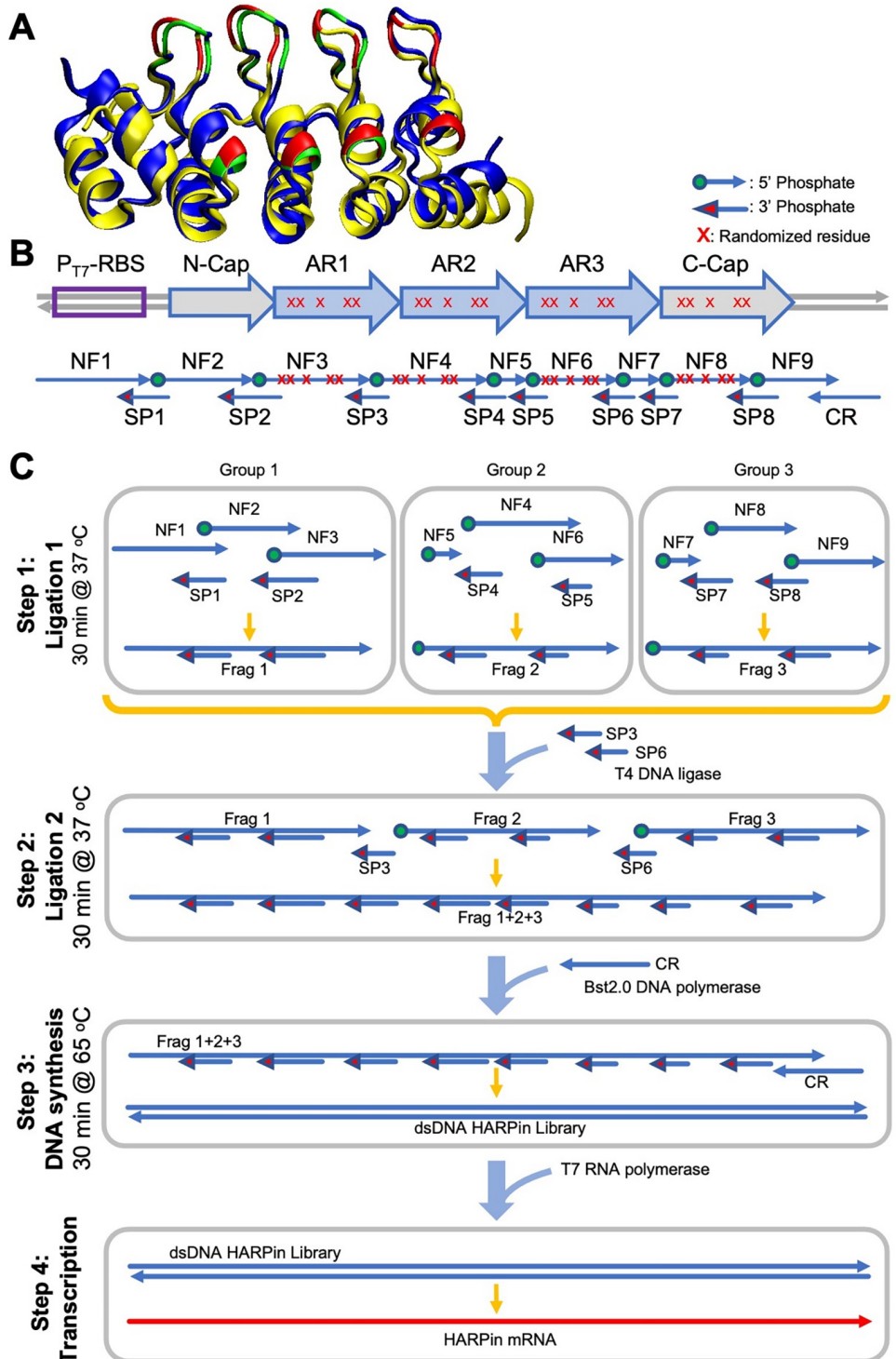

**Fig 1. Overview of Ligation & Elongation (L&E) library preparation method.** (A) Structural alignment of HARPin (pdb: 3v30, yellow) and DARPin (pdb: 4j8y, blue). Residues in red and green color are randomized in the HARPin and DARPin libraries, respectively. (B) Schematic of an HARPin library and the oligos. Red x's mark locations of randomized residues, which correspond to red residues in panel A. Green circles indicate the presence of 5' phosphate moiety; red arrows indicate 3' phosphate moiety. AR: ankyrin repeat. (C) L&E library construction workflow. The groups in Step 1 are as follows: Group 1: NF1, NF2, NF3, SP1, and SP2; Group 2: NF4, NF5, NF6, SP4, and SP5; and Group 3: NF7, NF8, NF9, SP7, SP8. NF: N-terminal forward, SP: Splint, CR: C-terminal reverse.

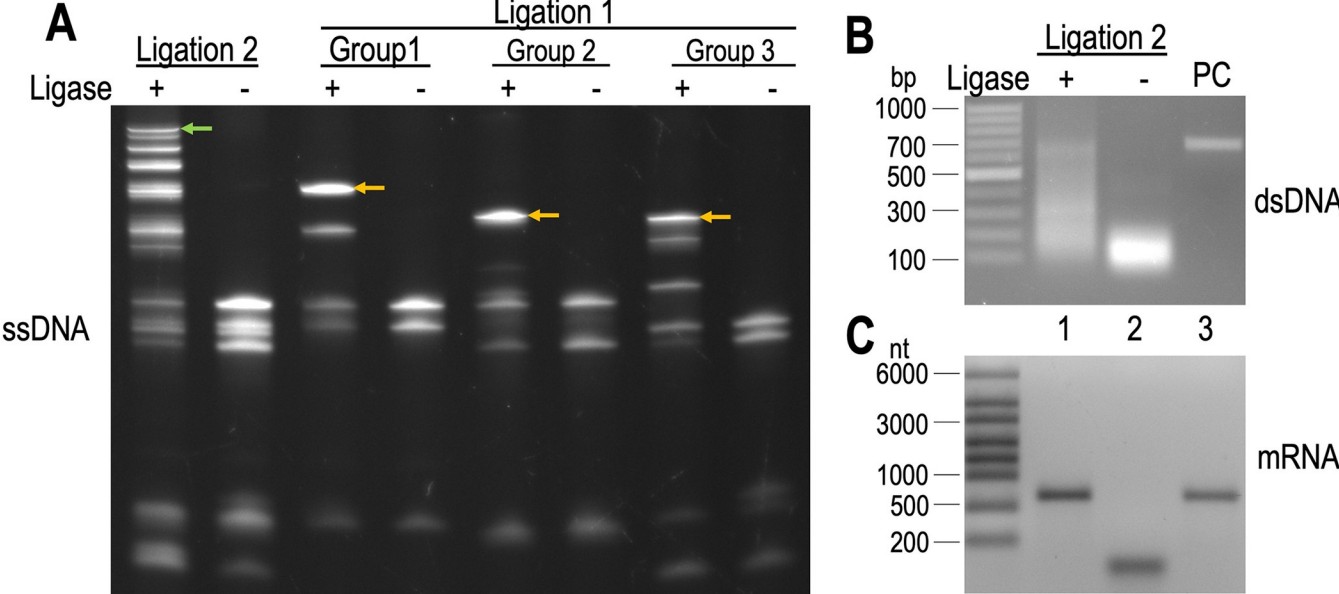

**Fig 2. Library preparation results.** (A) Single-stranded DNA products of Step 1 and 2 visualized on a 15% TBE-Urea gel stained with SYBR Green. Yellow arrows mark the three-way ligation products of Step 1, and the green arrow marks the presumed complete ligation product after combining all three groups. (B) Double-stranded DNA products from Step 3 visualized on a 1% agarose gel stained with ethidium bromide. Lanes 1 and 2 used DNA template of Ligation 2 (Step 2) produced in the presence and absence of ligase, respectively. Lane 3 (PC) contains gel-purified full-length HARPin double-stranded DNA (693 bp) amplified by PCR as a positive control. (C) mRNA product from Step 4 visualized on a denaturing 1.5% agarose gel stained with EtBr (1.5% agarose, 1% bleach). Lanes 1–3 contained the corresponding transcription product from panel B. The full-length mRNA contains 669 nt. The gels are representative of three independent experiments.

To demonstrate the utility of the DNA library without the need of purification, five microliters of the reaction product of Step 3 was used directly for mRNA transcription (Step 4). This *in vitro* transcription reaction routinely yielded 108–217 μg mRNA in 20 μL reaction volume. As anticipated, and despite the DNA template exhibiting a smearing pattern, the *in vitro* transcription reaction produced a single RNA product of the expected size (669 nt, Fig 2C, lane 1), validating our hypothesis that only the full-length gene contains the antisense DNA of the T7 promoter and that the nonspecific and shortened DNA fragments produced during library creation do not interfere with mRNA transcription. Thus, transcribed mRNA can be purified using a commercial RNA clean-up kit (e.g., Zymo Research EZNA RNA Clean & Concentrate kit) and be eluted in a buffer appropriate for the subsequent *in vitro* protein synthesis reaction.

The amount of the full-length double-stranded DNA product from Step 3 was quantified by qPCR (Fig 3). Starting from 15 pmoles (or $9 \times 10^{12}$ molecules) of each sense oligo in Step 1, we routinely obtained $4.25 \times 10^{11}$–$1.07 \times 10^{12}$ molecules of the double-stranded full-length gene product in the 150 μL reaction volume of Step 3, corresponding to an overall full-length gene ligation yield of 4.7–11.8%. Since the overall ligation efficiency is inversely proportional to the number of ligation events, increasing the length–thus reducing the number–of sense oligos may lead to a higher overall ligation efficiency. Importantly, the entire process of assembling the DNA library took one and a half hours and the reaction volume can be easily scaled up linearly for larger libraries. Moreover, it is conceivable that higher concentrations of oligos can be used during the ligation, making it possible to further reduce the reaction volume for larger libraries. This contrasts with the conventional PCR-based method used, for example, by Kondo, et al. which took months to create a monobody library with $10^{13}$ variants [13]. In addition, unlike the conventional PCR-based library creation methods that are at risk of introducing amplification bias and skewing the library diversity, our L&E method is PCR-free, offering

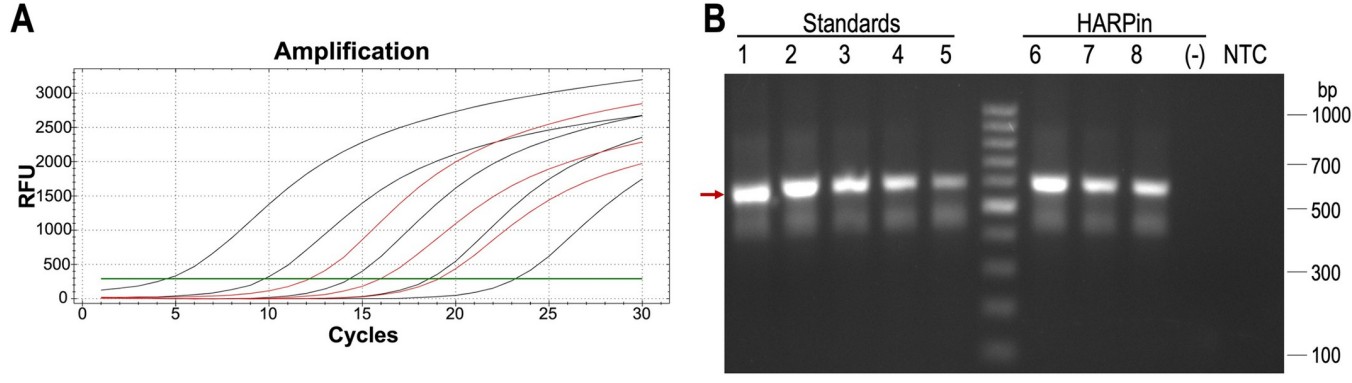

**Fig 3. L&E library quantification.** (A) qPCR amplification plot. Black lines represent the standard curve. Red lines represent the serially diluted library samples (1:50, 1:250, 1:1250). (B) Visualization of the final qPCR products on a 1% agarose gel stained with EtBr. Lanes 1–5 are from reactions containing $9x10^9$ – $9x10^5$ molecules of the standard DNA. Lanes 6–9 are from reactions containing 50-, 250- and 1250-fold diluted HARPin library samples. Red arrow indicates the desired amplicon (609 bp). (-): qPCR product from ligase-negative control reaction. NTC: No template control PCR. Middle lane contains GeneRuler 100 bp Plus DNA ladder (Thermo Fisher); the most intense band corresponds to 500 bp. (C) Starting quantity (SQ) calculation based on the qPCR standard curve. Values represent the copy number. Total SQ is the extrapolated total copy number of library in a final 150 μL double-stranded DNA synthesis reaction.

it the potential to most faithfully preserve the DNA library diversity afforded by chemical synthesis.

We employed Next-Generation Sequencing (NGS) to assess the quality of this new library. To simulate the conditions of the library's use in protein engineering, 500 nanograms of the *in vitro* transcription mRNA product (from Fig 1, Step 4) was reverse transcribed and then PCR amplified to append the NGS linkers. Approximately 750 nanograms of the PCR product was subjected to the MiSeq Next-Generation Sequencing reaction and yielded a total of 293,067 sequence reads (Fig 4A). After removing duplicate sequences and those of incorrect length, the cleaned dataset contained 162,422 unique reads (Fig 4A). The frequency of each base at the randomized codon position ranges between 19.3% to 33.7% for the N nucleotides and 38.4% to 61.1% for the K nucleotides (S1 Fig), which is consistent with oligos synthesized with machine mixing (Fig 4B and S1 Fig). The average frequency of each amino acid at the random-ized codon positions varies from 1.9% to 10.4%, which is consistent within that afforded by the NNK codon (Fig 4C and S2 Fig). Overall, 51.4% of the full-length reads contain at least one internal stop codon (Fig 4A). This is slightly lower than that predicted for a gene with 20 ran-domized NNK codons ($62.5\% = \frac{1}{32}$ STOP codon frequency at each position x 20 positions, assuming an even distribution of each possible base at the N and K positions). The final dataset of full-length sequences contained 78,808 unique variants (Fig 4A). The abundance of unique full-length sequences combined with a lack of noteworthy codon bias validate the use of L&E method for *in vitro* library creation.

In conclusion, we report a simple and efficient L&E method for synthetic library creation. A synthetic library with $\geq 10^{11}$ variants was produced in a 150 μL reaction volume. The entire

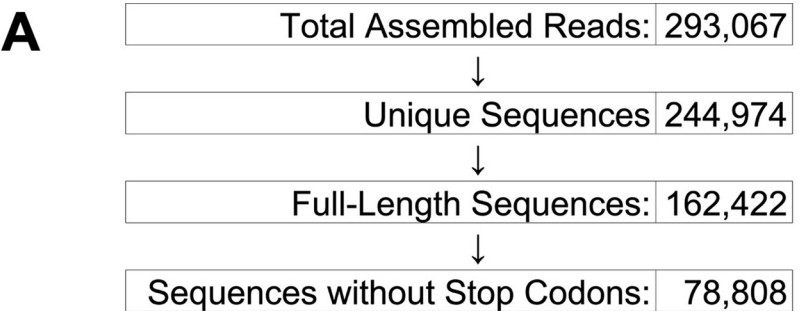

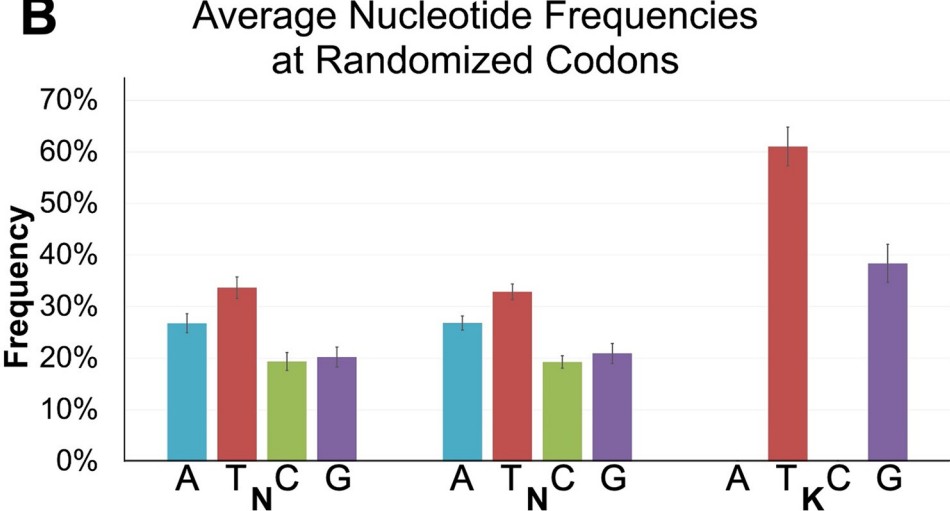

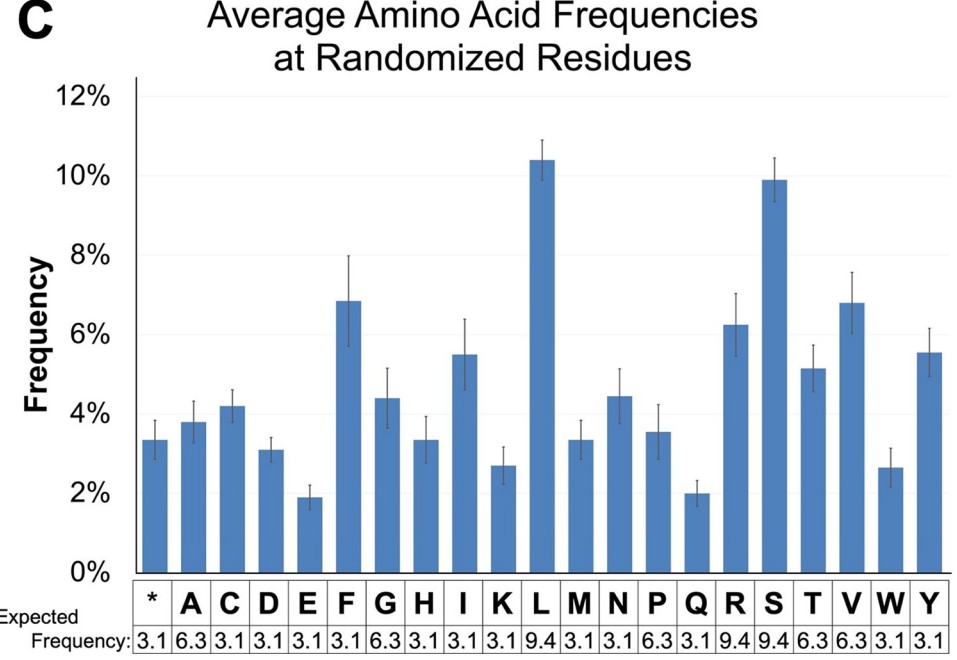

**Fig 4. Next-Generation Sequencing results.** (A) Results of sequencing reads at each step of analysis. (B, C) Frequency analysis of the dataset containing de-duplicated, full-length sequences. (B) Average nucleotide frequency at each position of the NNK randomized codons. (C) Average frequency of each amino acid at the randomized residues *:

STOP codon. Expected amino acid frequency denotes the relative frequency of each amino acid encoded by the 32 possible NNK codons, assuming an even distribution of possible nucleotides at each N and K position. Error bars represent standard deviations of the means. Expected frequency was calculated by dividing the number of codons for each amino acid by the total number of possible codons encoded by NNK (*i.e.*, 32).

process, from the commercially available oligonucleotides to a DNA template ready for mRNA transcription, took 1.5 hours. Furthermore, since the *in vitro* reaction volume is linearly scalable, more extensive libraries can be easily accommodated without increasing the reaction time. Finally, although only the HARPin scaffold was used here, we believe that our method should be extendable to any synthetic library, making L&E an alternative library creation method that has the potential to significantly accelerate the *in vitro* protein engineering process.

## Materials and methods

### Oligonucleotides

All oligonucleotides were synthesized by Integrated DNA Technologies (Coralville, IA; S1 Table) and PAGE purified in-house. Briefly, 5 μL of 100 μM stock of each oligo was loaded onto a 5% Urea-PAGE gel and visualized after staining with SYBR Green II nucleic acid stain (Thermo Fisher; Waltham, MA). The appropriate bands were excised and resuspended in 800 μL TE buffer (10 mM Tris-HCl pH 8.0, 0.1 mM EDTA), underwent three rounds of freeze-thaw to disrupt the gel, and incubated at 37˚C for 24 hours with shaking. The supernatant was transferred to fresh tubes the next day, and the DNA was concentrated via ethanol precipitation.

### Library preparation

The sense and splint oligos were divided into three groups. Group 1: NF1, NF2, NF3, SP1, and SP2; Group 2: NF4, NF5, NF6, SP4, and SP5; and Group 3: NF7, NF8, NF9, SP7, SP8. The HARPin library was created in three steps (Fig 1C). In Step 1, each group of oligos (15 pmol each) were mixed with 2 U of T4 DNA ligase (Lucigen; Madison, WI), Ribo-ATP (1 mM, Thermo Fisher), PEG-8000 (7.5%, VWR; Wayne, PA) in 30 μL of 1x NEB buffer r2.1 (NEB; Ipswich, MA) and incubated at 37˚C for 30 minutes. In Step 2, the three reactions from Step 1 were combined along with oligos SP3 and SP6 (15 pmoles each), 2 U of fresh T4 ligase, Ribo-ATP (final 1 mM), and PEG-8000 (final 7.5%) in 100 μL total reaction volume and incubated for an additional 30 minutes at 37˚C to generate single-stranded DNA sense molecules spanning the entire HARPin gene and the T7 promoter. Aliquots of reactions from Steps 1 and 2 were visualized on Novex 15% TBE-Urea gels (Thermo Fisher) after staining with SYBR Green II nucleic acid stain (Thermo Fisher).

In Step 3, to generate the full-length double-stranded DNA, primer CR (15 pmoles) was added to the reaction mixture together with dNTPs (final 1.4 mM each, NEB); 15 μL of 10x isothermal amplification buffer (NEB); MgSO$_4$ (final 8 mM total, NEB); and *Bst 2.0* WarmStart polymerase (48 units, NEB) in a final reaction volume of 150 μL. The reaction was carried out at 65˚C for 30 minutes, followed by heat inactivation at 80˚C for 20 minutes. The double-stranded DNA molecules generated were visualized on 1% agarose gels stained with ethidium bromide (EtBr). GeneRuler 100 bp Plus ladder (Thermo Fisher) was used for comparison.

### *In vitro* transcription

To demonstrate that only the full-length genes contain the anti-sense oligo of the promoter recognizable by the T7 RNA polymerase, 5 μL of the product from Step 3 was used directly for

mRNA transcription reaction (20 μL total volume) using the TranscriptAid T7 High Yield Transcription kit (Thermo Fisher). After a 1.5-hour incubation at 37°C, RNase-Free DNase I (2 U, NEB) and DNase I Buffer (NEB) were added to the mixture to remove the DNA template (50 μL final reaction volume). The transcription products were cleaned up using RNA Clean & Concentrator-25 kit (Zymo Research; Irvine, CA). An aliquot of the synthesized mRNA was visualized on denaturing agarose gels (1.5% agarose and 1% bleach [21]) stained with EtBr. RiboRuler RNA Ladder, High Range (Thermo Scientific) was used for size comparison.

### Library quantification

The amount of double stranded full-length HARPin gene synthesized in Step 3 was quantified via qPCR using primers HARPin_F and HARPin_R (S1 Table). A standard curve was constructed using serially diluted full-length HARPin DNA generated by PCR amplification and purified by agarose gel electrophoresis. Standard curve quantification results are shown in S2 Table. Three different dilutions of the library samples (Step 3) were used in the qPCR reactions to improve the estimation accuracy. All qPCR reactions were carried out as triplicates using Forget-Me-Not EvaGreen qPCR master mix (Biotium; Fremont, CA) and a Bio-Rad CFX96 Real-Time PCR system (Bio-Rad; Hercules, CA). The results were analyzed with Bio-Rad CFX Manager software. All qPCR reactions were also analyzed on 1% agarose gels to ensure that the quantification matches the desired target DNA product.

### Library diversity analysis

The diversity of our library was analyzed using next generation sequencing (NGS). First, 500 ng of the mRNA library produced in Step 4 (Fig 1C) was reverse transcribed using the SuperScript II reverse transcription kit (Invitrogen) according to the manufacturer's instruction, with oligo CR (S1 Table) serving as the primer. The resulting cDNA was then PCR amplified with Q5 High-Fidelity Polymerase (NEB) using primers NGS_F and NGS_R (S1 Table), which harbor the Illumina adapter sequences. The PCR product was then gel purified using the ZymoClean Gel DNA Recovery kit (Zymo Research) and subjected to Amplicon-EZ Next Generation Sequencing (Azenta Life Sciences; Chelmsford, MA). Paired reads were assembled using PEAR [22]. The dataset was processed (*i.e.*, de-duplicated and filtered to only include full-length sequences) using Geneious Prime (v.2022.1.1, Biomatters Ltd; San Diego, CA). Frequency grid profiles were constructed using Unipro UGENE [23]. Sequences including stop codons were removed from the dataset using a Biopython script (https://github.com/milesroberts-123/extract-weird-proteins).

### Supporting information

**S1 Fig. Nucleotide frequency analysis.** Nucleotide frequency at each position of the sequenced amplicon, including reference sequence. Nucleotide numbers refer to position within the complete HARPin gene. Randomized codons are depicted in red. Analysis performed on dataset containing de-duplicated, full-length sequences.
(PDF)

**S2 Fig. Amino acid frequency analysis.** Frequency of each amino acid at each position of the sequenced amplicon, including reference sequence. Residue numbers refer to position within the complete HARPin gene. Randomized residues are depicted in red. Analysis performed on dataset containing de-duplicated, full-length sequences.
(PDF)

**S1 Table. Primer sequences.**
(PDF)

**S2 Table. qPCR standard curve results.**
(PDF)

**S1 Graphical abstract.**
(TIF)

## Acknowledgments

Portions of this research were conducted with the advanced computing resources provided by Texas A&M High Performance Research Computing. Graphical abstract created with BioRender.com. We want to thank Benjamin Thomas for helpful discussion during the NGS data analysis.

## Author Contributions

**Conceptualization:** Zhilei Chen.

**Data curation:** Michael Woolley.

**Formal analysis:** Michael Woolley.

**Funding acquisition:** Zhilei Chen.

**Investigation:** Michael Woolley.

**Methodology:** Michael Woolley, Zhilei Chen.

**Project administration:** Zhilei Chen.

**Resources:** Zhilei Chen.

**Supervision:** Zhilei Chen.

**Validation:** Michael Woolley.

**Visualization:** Michael Woolley.

**Writing – original draft:** Michael Woolley.

**Writing – review & editing:** Zhilei Chen.

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
