## [Decision Letter · Decision Letter 0]

12 May 2022

PONE-D-22-08270

A PCR-free rapid protocol for one-pot construction of highly diverse libraries for in vitro protein engineering

PLOS ONE

Dear Dr. Chen,

Thank you for submitting your manuscript to PLOS ONE. After careful consideration, we feel that it has merit but does not fully meet PLOS ONE’s publication criteria as it currently stands. Therefore, we invite you to submit a revised version of the manuscript that addresses the points raised during the review process.

Your manuscript has been reviewed by three experts. Although they found it interesting, they think it necessary to incorporate additional data in the revised manuscript before publication. Please revise it according to their suggestions. Especially, sequencing data, preferably by NGS, to show diversity of the library should be necessary.

We look forward to receiving your revised manuscript.

Kind regards,

Hodaka Fujii, M.D., Ph.D.

Academic Editor

PLOS ONE

Journal Requirements:

4. We note you have included a table to which you do not refer in the text of your manuscript. Please ensure that you refer to Table 2 in your text; if accepted, production will need this reference to link the reader to the Table.

Reviewers' comments:

Reviewer's Responses to Questions

**Comments to the Author**

1. Is the manuscript technically sound, and do the data support the conclusions?

Reviewer #1: Yes

Reviewer #2: Partly

Reviewer #3: Yes

2. Has the statistical analysis been performed appropriately and rigorously? 

Reviewer #1: N/A

Reviewer #2: N/A

Reviewer #3: N/A

3. Have the authors made all data underlying the findings in their manuscript fully available?

Reviewer #1: Yes

Reviewer #2: Yes

Reviewer #3: Yes

4. Is the manuscript presented in an intelligible fashion and written in standard English?

Reviewer #1: Yes

Reviewer #2: Yes

Reviewer #3: Yes

5. Review Comments to the Author

Reviewer #1: The manuscript entitled "A PCR-free rapid protocol for one-pot construction of highly diverse libraries for in vitro protein engineering" by Woolley and Chen Claims a fast and simple mRNA library preparation method using ligation and elongation reaction. The authors utilized T4 DNA ligase to assemble and increase the size of the fragment with the aid of splint oligos. Bst2.0 polymerase then synthesized DNA from the CR oligo and strand-displaced the residual splint oligos to complete dsDNA. Simultaneous in vitro transcription could transcribe a sizeable mRNA library without further purification.

The method is clear, and the final result (mRNA library) is promising. I believe the authors can improve the manuscript with the following suggestions.

Major

1) It is unclear the actual diversity of the library by only showing the number of the library. Can you provide the sequencing data?

2) Due to the base-pairing between fragments and splint oligo, it seems you cannot randomize those regions (base-pairing regions). How many splint oligo bases do pair with fragments? Do you have a specific design role?

3) Figure 1, Steps 1-3: Residual SP oligos should be displayed in the figure because those SP oligos will not be strand-displaced until the end of step 3 by Bst2.0 polymerase.

Minor

1) Page 3, lines 24-30: Showing one or two directed evolution examples will strengthen the instruction part.

2) Page 5, lines 8-88: "was clearly visible on the gel" - How can you be sure this band is the regions for T7 promoter and RBS without comparing ladders? I was expecting the ladder in the original image in the supplementary information.

3) Page 7, lines 138-146: Although this manuscript focuses on the methodology but adding short comments on a potential ongoing protein engineering project in the conclusion part is highly recommended.

4) Page 8, line 151: Table 1 should be Table 2.

Recommend using the same table styles for Table 1 and Table 2.

Reviewer #2: The authors describe a ‘one-pot ligation and elongation’ method to accelerate the preparation of double-stranded DNA libraries for mRNA display of protein variants. mRNA display is a powerful technique, and the focus on library preparation should be of interest to the practitioner. That said, I do have reservations about publication of the work in current form (with the caveat that methods for DNA assembly and mRNA display are not my research focus). These fall into two categories:

1) Is the DNA assembly aspect of the work adequately placed in context? Some mention of the use of ‘splinted’ ligations in other situations may be helpful, for example:

Chen H, Weng J, Jiang K, Bao J (1990) A new method for the synthesis of a structural gene. Nucleic Acids Res. 18(4): 871-878.

-perhaps the original reference for assembly of ssDNA by enzymatic ligation using bridging oligos

Sui Z, An R, Komiyama M, Liang X (2021) Stepwise strategy for one-pot synthesis of single-stranded DNA rings from multiple short fragments. ChemBioChem. 22:1005-1011

-a recent application of one-pot, splinted ligation of ssDNA oligos to the preparation of circular ssDNA

2) Are sufficient data provided to establish the method as a sound one for mRNA display of protein libraries? Sequencing a subset of assembly products could provide an important quality control check for library construction. For example, the authors should sequence at least 20 library constructs to show the absence of insertions/deletions/mutations etc. at ligation points (i.e., that the construct remains in frame). Even better would be to use NGS (for example, do the amino acid frequencies of randomized positions match the expected frequencies, based on the library design?). Using an assembly product in translation would also be of interest, though is perhaps beyond the scope of this work.

Minor points:

-consider adding splint oligos to the Graphical Abstract, with unique coloring

-throughout, please define abbreviations (e.g. NF, CR, etc.)

-in the discussion, the authors may comment briefly on the specific steps involved to interface their library with mRNA display

-Fig 2b: The authors may clarify why a ‘smear’ is obtained during dsDNA synthesis, rather than a series of resolved co-products

-in Methods section, line 151: ‘Table 1’ should be ‘Table 2’

Reviewer #3: The authors report a one-pot method for the preparation of libraries starting from oligos pools.

For minor revisions, the authors should comment on the following:

- If full-length ligation yield is between ~5-12%, how is this expected to change as the number or length of sense oligos changes for different protein targets?

- The manuscript does not provide considerations beyond transcription, which raises some questions about utility:

-- While the method may be scalable, what are the scales tentatively required for protein engineering applications?

-- Is the fill-lenght mRNA yield negatively affected due to the lack of cleanup to alternate constructs?if so, to what extent? Figure 3c demonstrates significant significant quantity of 300-500bp and 700+bp products. The gel is cut off below ~100bp, and smaller truncated products are not shown to the reader.

-- is the one pot method compatible with in vitro translation? Or does the mRNA require cleanup/purification before translation?

-- For true utility with protein engineering, the compatibility with cell-free protein synthesis should be discussed or demonstrated.

-- is the scale used in this manuscript compatible with producing enough protein for analysis/characterization/selection?

6. PLOS authors have the option to publish the peer review history of their article (what does this mean?). If published, this will include your full peer review and any attached files.

Reviewer #1: No

Reviewer #2: **Yes: **Zachary P Gates

Reviewer #3: No

---

## [Decision Letter · Decision Letter 1]

6 Sep 2022

PONE-D-22-08270R1A PCR-free rapid protocol for one-pot construction of highly diverse libraries for in vitro protein engineeringPLOS ONE

Dear Dr. Chen,

Thank you for submitting your manuscript to PLOS ONE. After careful consideration, we feel that it has merit but does not fully meet PLOS ONE’s publication criteria as it currently stands. Therefore, we invite you to submit a revised version of the manuscript that addresses the points raised during the review process.

 Your revised manuscript was reviewed by the original three reviewers. Some of them still raised a couple of issues. Please consider if they are acceptable to you.

We look forward to receiving your revised manuscript.

Kind regards,

Hodaka Fujii, M.D., Ph.D.

Academic Editor

PLOS ONE

Journal Requirements:

Reviewers' comments:

Reviewer's Responses to Questions

**Comments to the Author**

1. If the authors have adequately addressed your comments raised in a previous round of review and you feel that this manuscript is now acceptable for publication, you may indicate that here to bypass the “Comments to the Author” section, enter your conflict of interest statement in the “Confidential to Editor” section, and submit your "Accept" recommendation.

Reviewer #1: All comments have been addressed

Reviewer #2: (No Response)

Reviewer #3: (No Response)

2. Is the manuscript technically sound, and do the data support the conclusions?

Reviewer #1: Yes

Reviewer #2: Yes

Reviewer #3: Partly

3. Has the statistical analysis been performed appropriately and rigorously? 

Reviewer #1: N/A

Reviewer #2: N/A

Reviewer #3: Yes

4. Have the authors made all data underlying the findings in their manuscript fully available?

Reviewer #1: Yes

Reviewer #2: Yes

Reviewer #3: Yes

5. Is the manuscript presented in an intelligible fashion and written in standard English?

Reviewer #1: Yes

Reviewer #2: Yes

Reviewer #3: Yes

6. Review Comments to the Author

Reviewer #1: The authors carefully addressed the reviewer's comment, and the revised manuscript has improved. The manuscript is recommended for publication.

Reviewer #2: This revision satisfactorily addresses all points raised during review.

One minor comment, for the authors’ consideration: In the new paragraph describing NGS results, consider stating concisely the reasoning behind how the NGS results validate the L&E method for library creation (e.g. with respect to both assembly product integrity, and library diversity).

Reviewer #3: Authors should consider modifying the article title to “PCR-free rapid protocol for one-pot construction of highly diverse mRNA libraries”, as this change would address most of my concerns.

- Since the mRNA would require cleanup prior to in vitro translation, the protocol for protein engineering is no longer “one-pot”.

- Its true that 217 ug or ~1nmol mRNA in 20 uL reaction may be useful for mRNA display, the corresponding in vitro translation reaction is unlikely to produce enough protein for analysis or characterization, and only sufficient for selection in limited cases.

Therefore, the authors should demonstrate protein production and engineering efforts if that part of their claims.

Plugging the HARPin Library into NEB’s PURExpress kit could be a good starting point. Though this would also be inconsistent with the “one-pot” claim. Such an approach could be superior to purifying the mRNA prior to in vitro translation since handling of mRNA could cause problems and variability in different hands.

7. PLOS authors have the option to publish the peer review history of their article (what does this mean?). If published, this will include your full peer review and any attached files.

Reviewer #1: No

Reviewer #2: **Yes: **Zachary P Gates

Reviewer #3: No

---

## [Author Response · Author response to Decision Letter 1]

16 Sep 2022

Please see attached Response to Reviewers document.

---

## [Decision Letter · Decision Letter 2]

2 Oct 2022

PONE-D-22-08270R2A PCR-free rapid protocol for one-pot construction of highly diverse genetic libraries for in vitro protein engineeringPLOS ONE

Dear Dr. Chen,

Thank you for submitting your manuscript to PLOS ONE. After careful consideration, we feel that it has merit but does not fully meet PLOS ONE’s publication criteria as it currently stands. Therefore, we invite you to submit a revised version of the manuscript that addresses the points raised during the review process.

 One of the reviewers still raised a concern about the title of the manuscript. Could you consider changing it according to their suggestion?

We look forward to receiving your revised manuscript.

Kind regards,

Hodaka Fujii, M.D., Ph.D.

Academic Editor

PLOS ONE

Journal Requirements:

Reviewers' comments:

Reviewer's Responses to Questions

**Comments to the Author**

1. If the authors have adequately addressed your comments raised in a previous round of review and you feel that this manuscript is now acceptable for publication, you may indicate that here to bypass the “Comments to the Author” section, enter your conflict of interest statement in the “Confidential to Editor” section, and submit your "Accept" recommendation.

Reviewer #1: All comments have been addressed

Reviewer #2: All comments have been addressed

Reviewer #3: (No Response)

2. Is the manuscript technically sound, and do the data support the conclusions?

Reviewer #1: Yes

Reviewer #2: Yes

Reviewer #3: (No Response)

3. Has the statistical analysis been performed appropriately and rigorously? 

Reviewer #1: Yes

Reviewer #2: N/A

Reviewer #3: (No Response)

4. Have the authors made all data underlying the findings in their manuscript fully available?

Reviewer #1: Yes

Reviewer #2: Yes

Reviewer #3: (No Response)

5. Is the manuscript presented in an intelligible fashion and written in standard English?

Reviewer #1: Yes

Reviewer #2: Yes

Reviewer #3: (No Response)

6. Review Comments to the Author

Reviewer #1: (No Response)

Reviewer #2: This revision satisfactorily addresses all points raised during review, and is recommended for publication.

Reviewer #3: Authors state that:

"Based on the reviewer’s suggestion, we modified the title of the manuscript as follows":

'A PCR-free rapid protocol for one-pot construction of highly diverse genetic libraries for in vitro

protein engineering' ".

This title appears to be unchanged and does not resolve my concerns.

To reiterate, the authors have not demonstrated that their method can be used for in vitro protein engineering, or that protein engineering can be done in "one-pot".

Premise of protein engineering should be removed form the title, possible options:

'A PCR-free rapid protocol for one-pot construction of highly diverse genetic libraries'

'A PCR-free rapid protocol for one-pot construction of highly diverse mRNA libraries'

7. PLOS authors have the option to publish the peer review history of their article (what does this mean?). If published, this will include your full peer review and any attached files.

Reviewer #1: No

Reviewer #2: **Yes: **Zachary P Gates

Reviewer #3: No

---

## [Author Response · Author response to Decision Letter 2]

3 Oct 2022

Please see attached "Response to Reviewers" document.

---

## [Editor Report · Decision Letter 3]

5 Oct 2022

A PCR-free rapid protocol for one-pot construction of highly diverse genetic libraries

PONE-D-22-08270R3

Dear Dr. Chen,

We’re pleased to inform you that your manuscript has been judged scientifically suitable for publication and will be formally accepted for publication once it meets all outstanding technical requirements.

Kind regards,

Hodaka Fujii, M.D., Ph.D.

Academic Editor

PLOS ONE
---

## [Editor Report · Acceptance letter]

21 Oct 2022

PONE-D-22-08270R3 

A PCR-free rapid protocol for one-pot construction of highly diverse genetic libraries 

Dear Dr. Chen:

I'm pleased to inform you that your manuscript has been deemed suitable for publication in PLOS ONE. Congratulations! Your manuscript is now with our production department. 

Kind regards, 

on behalf of

Dr. Hodaka Fujii 

Academic Editor

PLOS ONE